# CERTIFIED TRAINING:
# SMALL BOXES ARE ALL YOU NEED

**Mark Niklas Müller**,[*] **Franziska Eckert**,[*] **Marc Fischer & Martin Vechev**
Department of Computer Science
ETH Zurich, Switzerland
{mark.mueller, marc.fischer, martin.vechev}@inf.ethz.ch, eckertf@student.ethz.ch

## ABSTRACT

To obtain, deterministic guarantees of adversarial robustness, specialized training methods are used. We propose, SABR, a novel such certified training method, based on the key insight that propagating interval bounds for a small but carefully selected subset of the adversarial input region is sufficient to approximate the worst-case loss over the whole region while significantly reducing approximation errors. We show in an extensive empirical evaluation that SABR outperforms existing certified defenses in terms of both *standard and certifiable accuracies* across perturbation magnitudes and datasets, pointing to a new class of certified training methods promising to alleviate the robustness-accuracy trade-off.

## 1 INTRODUCTION

As neural networks are increasingly deployed in safety-critical domains, formal robustness guarantees against adversarial examples (Biggio et al., 2013; Szegedy et al., 2014) are becoming ever more important. However, despite significant progress, specialized training methods that improve certifiability at the cost of severely reduced accuracies are still required to obtain deterministic guarantees.

Given an input region defined by an adversary specification, both training and certification methods compute a network's reachable set by propagating a symbolic over-approximation of this region through the network (Singh et al., 2018; 2019a; Gowal et al., 2018a). Depending on the propagation method, both the computational complexity and approximation-tightness can vary widely. For certified training, an over-approximation of the worst-case loss is computed from this reachable set and then optimized (Mirman et al., 2018; Wong et al., 2018). Surprisingly, the least precise propagation methods yield the highest certified accuracies as more precise methods induce harder optimization problems (Jovanovic et al., 2021). However, the large approximation errors incurred by these imprecise methods lead to over-regularization and thus poor accuracy. Combining precise worst-case loss approximations and a tractable optimization problem is thus the core challenge of certified training.

In this work, we tackle this challenge and propose a novel certified training method, SABR, **S**mall **A**dversarial **B**ounding **R**egions, based on the following key insight: by propagating small but carefully selected subsets of the adversarial input region with imprecise methods (i.e., BOX), we can obtain *both* well-behaved optimization problems and precise approximations of the worst-case loss. This yields less over-regularized networks, allowing SABR to improve on state-of-the-art certified defenses in terms of both standard *and* certified accuracies across settings, thereby pointing to a new class of certified training methods.

**Main Contributions**   Our main contributions are:

- A novel certified training method, SABR, reducing over-regularization to improve both standard and certified accuracy (§3).
- A theoretical investigation motivating SABR by deriving new insights into the growth of BOX relaxations during propagation (§4).
- An extensive empirical evaluation demonstrating that SABR outperforms *all* state-of-the-art certified training methods in terms of both *standard and certifiable accuracies* on MNIST, CIFAR-10, and TINYIMAGENET (§5).

---

[*]Equal contribution

## 2 BACKGROUND

In this section, we provide the necessary background for SABR.

**Adversarial Robustness**  Consider a classification model $h\colon \mathbb{R}^{d_{\text{in}}} \mapsto \mathbb{R}^c$ that, given an input $x \in \mathcal{X} \subseteq \mathbb{R}^{d_{\text{in}}}$, predicts numerical scores $y := h(x)$ for every class. We say that $h$ is adversarially robust on an $\ell_p$-norm ball $\mathcal{B}_p^{\epsilon_p}(x)$ of radius $\epsilon_p$ if it consistently predicts the target class $t$ for all perturbed inputs $x' \in \mathcal{B}_p^{\epsilon_p}(x)$. More formally, we define *adversarial robustness* as:

$$\arg\max_j h(x')_j = t, \quad \forall x' \in \mathcal{B}_p^{\epsilon_p}(x) := \{x' \in \mathcal{X} \mid \|x - x'\|_p \leq \epsilon_p\}. \tag{1}$$

**Neural Network Verification**  To verify that a neural network $h$ is adversarially robust, several verification techniques have been proposed.

A simple but effective such method is verification with the BOX relaxation (Mirman et al., 2018), also called interval bound propagation (IBP) (Gowal et al., 2018b). Conceptually, we first compute an over-approximation of a network's reachable set by propagating the input region $\mathcal{B}_p^{\epsilon_p}(x)$ through the neural network and then check whether all outputs in the reachable set yield the correct classification. This propagation sequentially computes a hyper-box (each dimension is described as an interval) relaxation of a layer's output, given a hyper-box input. As an example, consider an $L$-layer network $h = f_L \circ \sigma \circ f_{L-2} \circ \ldots \circ f_1$, with linear layers $f_i$ and ReLU activation functions $\sigma$. Given an input region $\mathcal{B}_p^{\epsilon_p}(x)$, we over-approximate it as a hyper-box, centered at $\bar{x}^0 := x$ and with radius $\delta^0 := \epsilon_p$,

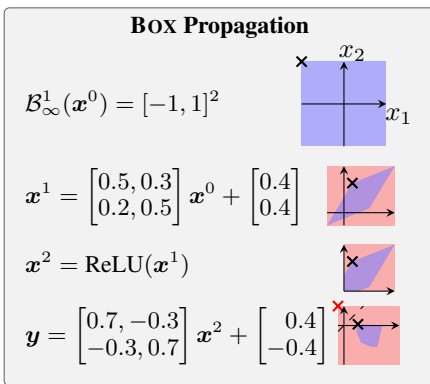

BOX Propagation

$$\mathcal{B}_\infty^1(x^0) = [-1, 1]^2$$

$$x^1 = \begin{bmatrix} 0.5, 0.3 \\ 0.2, 0.5 \end{bmatrix} x^0 + \begin{bmatrix} 0.4 \\ 0.4 \end{bmatrix}$$

$$x^2 = \text{ReLU}(x^1)$$

$$y = \begin{bmatrix} 0.7, -0.3 \\ -0.3, 0.7 \end{bmatrix} x^2 + \begin{bmatrix} 0.4 \\ -0.4 \end{bmatrix}$$

Figure 1: Comparison of exact (blue) and BOX (red) propagation through a one layer network. We show the concrete points maximizing the logit difference $y_2 - y_1$ as a black $\times$ and the corresponding relaxation as a red $\times$.

such that we have the $i^{\text{th}}$ dimension of the input $x_i^0 \in [\bar{x}_i^0 - \delta_i^0, \bar{x}_i^0 + \delta_i^0]$. Given a linear layer $f_i(x^{i-1}) = W x^{i-1} + b =: x^i$, we obtain the hyper-box relaxation of its output with centre $\bar{x}^i = W \bar{x}^{i-1} + b$ and radius $\delta^i = |W| \delta^{i-1}$, where $|\cdot|$ denotes the elementwise absolute value. A ReLU activation $\text{ReLU}(x^{i-1}) := \max(0, x^{i-1})$ can be over-approximated by propagating the lower and upper bound separately, resulting in a output hyper-box with $\bar{x}^i = \frac{u^i + l^i}{2}$ and $\delta^i = \frac{u^i - l^i}{2}$ where $l^i = \text{ReLU}(\bar{x}^{i-1} - \delta^{i-1})$ and $u^i = \text{ReLU}(\bar{x}^{i-1} + \delta^{i-1})$. Proceeding this way for all layers, we obtain lower and upper bounds on the network output $y$ and can check if the output score of the target class is greater than that of all other classes by computing the upper bound on the logit difference $y_i^\Delta := y_i - y_t$ and then checking whether $y_i^\Delta < 0, \ \forall i \neq t$.

We illustrate this propagation process for a one-layer network in Fig. 1. There, the blue shapes (■) show an exact propagation of the input region and the red shapes (■) their hyper-box relaxation. Note how after the first linear and ReLU layer (third row), the relaxation (red) contains already many points not reachable via exact propagation (blue), despite it being the smallest hyper-box containing the exact region. These so-called approximation errors accumulate quickly, leading to an increasingly imprecise abstraction, as can be seen by comparing the two shapes after an additional linear layer (last row). To verify that this network classifies all inputs in $[-1, 1]^2$ to class 1, we have to show the upper bound of the logit difference $y_2 - y_1$ to be less than 0. While the concrete maximum of $-0.3 \geq y_2 - y_1$ (black $\times$) is indeed less than 0, showing that the network is robust, the BOX relaxation only yields $0.6 \geq y_2 - y_1$ (red $\times$) and is thus too imprecise to prove it.

Beyond BOX, more precise verification approaches track more relational information at the cost of increased computational complexity (Palma et al., 2022; Wang et al., 2021). A recent example is MN-BAB (Ferrari et al., 2022), which improves on BOX in two key ways: First, instead of propagating axis-aligned hyper-boxes, it uses much more expressive polyhedra, allowing linear layers to be captured exactly and ReLU layers much more precisely. Second, if the result is still too imprecise, the verification problem is recursively split into easier ones, by introducing a case distinction between the two linear segments of the ReLU function. This is called the branch-and-bound (BaB) approach (Bunel et al., 2020). We refer the interested reader to Ferrari et al. (2022) for more details.

**Training for Robustness** For neural networks to be certifiably robust, special training is necessary. Given a data distribution $(\boldsymbol{x}, t) \sim \mathcal{D}$, standard training generally aims to find a network parametrization $\boldsymbol{\theta}$ that minimizes the expected cross-entropy loss (see App. B.1):

$$\theta_{\text{std}} = \arg\min_{\boldsymbol{\theta}} \mathbb{E}_{\mathcal{D}}[\mathcal{L}_{\text{CE}}(\boldsymbol{h}_{\boldsymbol{\theta}}(\boldsymbol{x}), t)], \quad \text{with} \quad \mathcal{L}_{\text{CE}}(\boldsymbol{y}, t) = \ln\big(1 + \sum_{i \neq t} \exp(y_i - y_t)\big). \quad (2)$$

When training for robustness, we, instead, wish to minimize the expected *worst-case loss* around the data distribution, leading to the min-max optimization problem:

$$\theta_{\text{rob}} = \arg\min_{\boldsymbol{\theta}} \mathbb{E}_{\mathcal{D}}\Big[\max_{\boldsymbol{x}' \in \mathcal{B}_p^{\epsilon_p}(\boldsymbol{x})} \mathcal{L}_{\text{CE}}(\boldsymbol{h}_{\boldsymbol{\theta}}(\boldsymbol{x}'), t)\Big]. \quad (3)$$

Unfortunately, solving the inner maximization problem is generally intractable. Therefore, it is commonly under- or over-approximated, yielding adversarial and certified training, respectively. For notational clarity, we henceforth drop the subscript $p$.

**Adversarial Training** Adversarial training optimizes a lower bound on the inner optimization objective in Eq. (3) by first computing concrete examples $\boldsymbol{x}' \in \mathcal{B}^{\epsilon}(\boldsymbol{x})$ maximizing the loss term and then optimizing the network parameters $\boldsymbol{\theta}$ for these samples. Typically, $\boldsymbol{x}'$ is computed by initializing $\boldsymbol{x}'_0$ uniformly at random in $\mathcal{B}^{\epsilon}(\boldsymbol{x})$ and then updating it over $N$ projected gradient descent steps (PGD) (Madry et al., 2018) $\boldsymbol{x}'_{n+1} = \Pi_{\mathcal{B}^{\epsilon}(\boldsymbol{x})} \boldsymbol{x}'_n + \alpha \, \text{sign}(\nabla_{\boldsymbol{x}'_n} \mathcal{L}_{\text{CE}}(\boldsymbol{h}_{\boldsymbol{\theta}}(\boldsymbol{x}'_n), t))$, with step size $\alpha$ and projection operator $\Pi$. While networks trained this way typically exhibit good empirical robustness, they remain hard to formally verify and sometimes vulnerable to stronger or different attacks (Tramèr et al., 2020; Croce & Hein, 2020).

**Certified Training** Certified training optimizes an upper bound on the inner maximization objective in Eq. (3), obtained via a bound propagation method. These methods compute an upper bound $\boldsymbol{u}_{\boldsymbol{y}^{\triangle}}$ on the logit differences $\boldsymbol{y}^{\triangle} := \boldsymbol{y} - y_t$, as described above, to obtain the robust cross-entropy loss $\mathcal{L}_{\text{CE,rob}}(\mathcal{B}^{\epsilon}(\boldsymbol{x}), t) = \mathcal{L}_{\text{CE}}(\boldsymbol{u}_{\boldsymbol{y}^{\triangle}}, t)$. We will use BOX to refer to the verification and propagation approach, and IBP to refer to the corresponding training method.

Surprisingly, using the imprecise BOX relaxation (Mirman et al., 2018; Gowal et al., 2018b; Shi et al., 2021) consistently produces better results than methods based on tighter abstractions (Zhang et al., 2020; Balunovic & Vechev, 2020; Wong et al., 2018). Jovanovic et al. (2021) trace this back to the optimization problems induced by the more precise methods becoming intractable to solve. While the heavily regularized, IBP trained networks are amenable to certification, they suffer from severely reduced (standard) accuracies. Overcoming this robustness-accuracy trade-off remains a key challenge of robust machine learning.

## 3 METHOD – SMALL REGIONS FOR CERTIFIED TRAINING

To train networks that are not only robust and amenable to certification but also retain comparatively high standard accuracies, we propose the novel certified training method, SABR — **S**mall **A**dversarial **B**ounding **R**egions. We leverage the key insight that computing an over-approximation of the worst-case loss over a small but carefully selected subset of the input region $\mathcal{B}^{\epsilon}(\boldsymbol{x})$ often yields a good proxy for the worst-case loss over the whole region while significantly reducing approximation errors.

We illustrate this intuition in Fig. 2. Existing certified training methods always consider the whole input region (dashed box ⊡ in the input panel). Propagating such large regions through the network yields quickly growing approximation errors and thus very imprecise over-approximations of the actual worst-case loss (compare the reachable set in red ■ and green ■ to the dashed box ⊡ in the output panel), causing significant over-regularization (large blue arrow ⬇). Adversarial training methods, in contrast, only consider individual points in the input space (× in Fig. 2) and often fail to capture the actual worst-case loss. This leads to insufficient regularization (small blue arrow ↓ in the output panel) and yields networks which are not amenable to certification and potentially not robust.

We tackle this problem by propagating small, adversarially chosen subsets of the input region (solid box □ in the input panel of Fig. 2), which we call the *propagation region*. This leads to significantly reduced approximation errors (see the solid box □ in the output panel) inducing a level of regularization in-between certified and adversarial training methods (medium blue arrow ⬇), allowing us to train networks that are both robust and accurate.

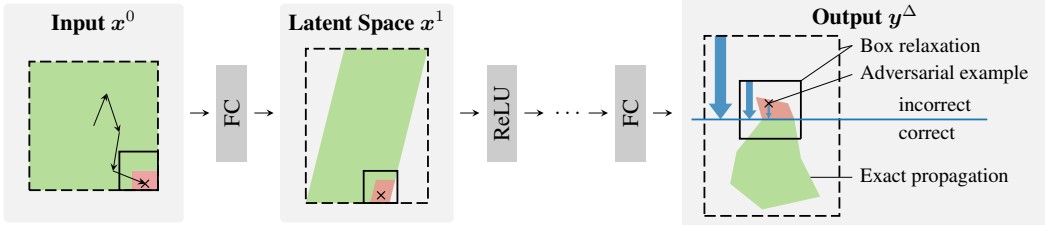

Figure 2: Illustration of SABR training. Instead of propagating a BOX approximation (dashed box ⬚) of the whole input region (red ▪ and green ▪ shapes in input space), SABR propagates a small subset of this region (solid box □), selected to contain the adversarial example (black ×) and thus the misclassified region (▪ red). The smaller BOX accumulates much fewer approximation errors during propagation, leading to a significantly smaller output relaxation, which induces much less regularization (medium blue ⬇) than training with the full region (large blue ⬇), but more than training with just the adversarial example (small blue ⬇).

More formally, we define an auxiliary objective for the robust optimization problem Eq. (3) as

$$\mathcal{L}_{\text{SABR}} = \max_{\boldsymbol{x}^* \in \mathcal{B}^\tau(\boldsymbol{x}')} \mathcal{L}_{\text{CE}}(\boldsymbol{x}^*, t), \qquad (4)$$

where we replace the maximum over the whole input region $\mathcal{B}^\epsilon(\boldsymbol{x})$ with that over a carefully selected subset $\mathcal{B}^\tau(\boldsymbol{x}')$. While choosing $\boldsymbol{x}' = \Pi_{\mathcal{B}^{\epsilon-\tau}(\boldsymbol{x})} \arg\max_{\boldsymbol{x}^* \in \mathcal{B}^\epsilon(\boldsymbol{x})} \mathcal{L}_{\text{CE}}(\boldsymbol{x}^*, t)$ would recover the original robust training problem (Eq. (3)), both, computing the maximum loss over a given input region (Eq. (4)) and finding a point that realizes this loss is generally intractable. Instead, we instantiate SABR by combining different approximate approaches for the two key components: a) a method for choosing the location $\boldsymbol{x}'$ and size $\tau$ of the propagation region, and b) a method used for propagating the thus selected region. Note that we thus generally do not obtain a sound over-approximation of the loss on $\mathcal{B}^\epsilon(\boldsymbol{x})$. Depending on the size of the propagated region $\mathcal{B}^\tau(\boldsymbol{x}')$, SABR can be seen as a continuous interpolation between adversarial training for infinitesimally small regions $\tau = 0$ and standard certified training for the full input region $\tau = \epsilon$.

**Selecting the Propagation Region**   SABR aims to find and propagate a small subset of the adversarial input region $\mathcal{B}^\epsilon(\boldsymbol{x})$ that contains the inputs leading to the worst-case loss. To this end, we parametrize this propagation region as an $\ell_p$-norm ball $\mathcal{B}^\tau(\boldsymbol{x}')$ with centre $\boldsymbol{x}'$ and radius $\tau \le \epsilon - \|\boldsymbol{x} - \boldsymbol{x}'\|_p$. We first choose $\tau = \lambda\epsilon$ by scaling the original perturbation radius $\epsilon$ with the subselection ratio $\lambda \in (0, 1]$. We then select $\boldsymbol{x}'$ as follows: We conduct a PGD attack, choosing the preliminary centre $\boldsymbol{x}^*$ as the sample with the highest loss. We then ensure the obtained region is fully contained in the original one by projecting $\boldsymbol{x}^*$ onto $\mathcal{B}^{\epsilon-\tau}(\boldsymbol{x})$ to obtain $\boldsymbol{x}'$. We show this in Fig. 3.

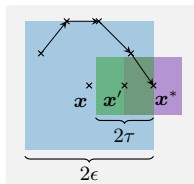

Figure 3: Illustration of propagation region selection process.

**Propagation Method**   Having found the propagation region $\mathcal{B}^\tau(\boldsymbol{x}')$, we can use any symbolic propagation method to compute an over-approximation of its worst-case loss. We chose BOX propagation (DIFFAI Mirman et al. (2018) or IBP (Gowal et al., 2018b)) to obtain well-behaved optimization problems (Jovanovic et al., 2021). There, choosing small propagation regions ($\tau \ll 1$), can significantly reduce the incurred over-approximation errors, as we will show later (see §4).

## 4   UNDERSTANDING SABR: ROBUST LOSS AND GROWTH OF SMALL BOXES

In this section, we aim to uncover the reasons behind SABR's success. Towards this, we first analyse the relationship between robust loss and over-approximation size before investigating the growth of the BOX approximation with propagation region size.

**Robust Loss Analysis**   Certified training typically optimizes an over-approximation of the worst-case cross-entropy loss $\mathcal{L}_{\text{CE,rob}}$, computed via the softmax of the upper-bound on the logit differences $\boldsymbol{y}^\Delta := \boldsymbol{y} - y_t$. When training with the BOX relaxation and assuming the target class $t = 1$, w.l.o.g., we obtain the logit difference $\boldsymbol{y}^\Delta \in [\bar{\boldsymbol{y}}^\Delta - \boldsymbol{\delta}^\Delta, \bar{\boldsymbol{y}}^\Delta + \boldsymbol{\delta}^\Delta]$ and thus the robust cross entropy loss

$$\mathcal{L}_{\text{CE, rob}}(\boldsymbol{x}) = \ln \left(1 + \sum_{i=2}^{n} e^{\bar{y}_i^\Delta + \delta_i^\Delta}\right). \qquad (5)$$

We observe that samples with high ($>0$) worst-case misclassification margin $\bar{y}^{\Delta}+\delta^{\Delta} := \max_i \bar{y}_i^{\Delta} + \delta_i^{\Delta}$ dominate the overall loss and permit the per-sample loss term to be approximated as

$$\max_i \bar{y}_i^{\Delta} + \delta_i^{\Delta} =: \bar{y}^{\Delta} + \delta^{\Delta} < \mathcal{L}_{\text{CE, rob}} < \ln(n) + \max_i \bar{y}_i^{\Delta} + \delta_i^{\Delta}. \tag{6}$$

Further, we note that the BOX relaxations of many functions preserve the box centres, i.e., $\bar{x}^i = f(\bar{x}^{i-1})$. Only unstable ReLUs, i.e., ReLUs containing 0 in their input bounds, introduce a slight shift. However, these are empirically few in certifiably trained networks (see Table 4).

These observations allow us to decompose the robust loss into an accuracy term $\bar{y}^{\Delta}$, corresponding to the misclassification margin of the adversarial example $x'$ at the centre of the propagation region, and a robustness term $\delta^{\Delta}$, bounding the difference to the actual worst-case loss. These terms generally represent conflicting objectives, as local robustness requires the network to disregard high frequency features (Ilyas et al., 2019). Therefore, robustness and accuracy are balanced to minimize the optimization objective Eq. (5). Consequently, reducing the regularization induced by the robustness term will bias the optimization process towards standard accuracy. Next, we investigate how SABR reduces exactly this regularization strength, by propagating smaller regions.

**Box Growth**   To investigate how BOX approximations grow as they are propagated, let us again consider an $L$-layer network $h = f_L \circ \sigma \circ f_{L-2} \circ \ldots \circ f_1$, with linear layers $f_i$ and ReLU activation functions $\sigma$. Given a BOX input with radius $\delta^{i-1}$ and centre distribution $\bar{x}^{i-1} \sim \mathcal{D}$, we now define the per-layer growth rate $\kappa^i$ as the ratio of input and expected output radius:

$$\kappa^i = \frac{\mathbb{E}_{\mathcal{D}}[\delta^i]}{\delta^{i-1}}. \tag{7}$$

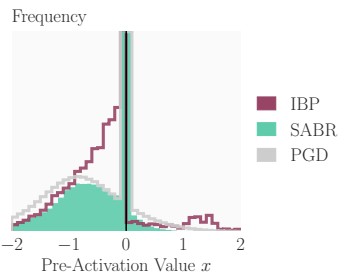

Figure 4: Input distribution for last ReLU layer depending on training method.

For linear layers with weight matrix $W$, we obtain an output radius $\delta^i = |W| \delta^{i-1}$ and thus a constant growth rate $\kappa^i$, corresponding to the row-wise $\ell_1$ norm of the weight matrix $|W_{j,\cdot}|_1$. Empirically, we find most linear and convolutional layers to exhibit growth rates between 10 and 100 (see Table 9 in App. D.4).

For ReLU layers $x^i = \sigma(x^{i-1})$, computing the growth rate is more challenging, as it depends on the location and size of the inputs. Shi et al. (2021) assume the input BOX centres $\bar{x}^{i-1}$ to be symmetrically distributed around 0, i.e., $P_{\mathcal{D}}(\bar{x}^{i-1}) = P_{\mathcal{D}}(-\bar{x}^{i-1})$, and obtain a constant growth rate of $\kappa^i = 0.5$. While this assumption holds at initialization, we observe that trained networks tend to have more inactive than active ReLUs (see Table 4), indicating asymmetric distributions with more negative inputs (see Fig. 4).

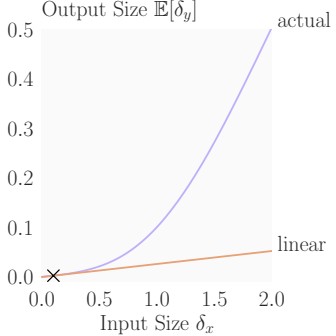

Figure 5: Comparison of the actual (purple) mean output size and a linear growth (orange) around the black $\times$ for a ReLU layer where input box centres $\bar{x} \sim \mathcal{N}(\mu = -1.0, \sigma = 0.5)$.

We now investigate this more realistic setting. We first consider the two limit cases where input radii $\delta^{i-1}$ go against 0 and $\infty$. When input radii are $\delta^{i-1} \approx 0$, active neurons will stay stably active, yielding $\delta^i = \delta^{i-1}$ and inactive neurons will stay stably inactive, yielding $\delta^i = 0$. Thus, we obtain a growth rate, equivalent to the portion of active neurons. In the other extreme $\delta^{i-1} \to \infty$, all neurons will become unstable with $\bar{x}^{i-1} \ll \delta^{i-1}$, yielding $\delta^i \approx 0.5 \, \delta^{i-1}$, and thus a constant growth rate of $\kappa^i = 0.5$. To analyze the behavior in between those extremes, we assume pointwise asymmetry favouring negative inputs, i.e., $p(\bar{x}^{i-1} = -z) > p(\bar{x}^{i-1} = z)$, $\forall z \in \mathbb{R}^{>0}$. In this setting, we find that output radii grow strictly super-linear in the input size:

**Theorem 4.1** (Hyper-Box Growth). *Let* $y := \sigma(x) = \max(0, x)$ *be a ReLU function and consider box inputs with radius* $\delta_x$ *and asymmetrically distributed centres* $\bar{x} \sim \mathcal{D}$ *such that* $P_{\mathcal{D}}(\bar{x} = -z) > P_{\mathcal{D}}(\bar{x} = z)$, $\forall z \in \mathbb{R}^{>0}$. *Then, the mean output radius* $\delta_y$ *will grow super-linearly in the input radius* $\delta_x$. *More formally:*

$$\forall \delta_x, \delta_x' \in \mathbb{R}^{\geq 0}: \quad \delta_x' > \delta_x \implies \mathbb{E}_{\mathcal{D}}[\delta_y'] > \mathbb{E}_{\mathcal{D}}[\delta_y] + (\delta_x' - \delta_x) \frac{\partial}{\partial \delta_x} \mathbb{E}_{\mathcal{D}}[\delta_y]. \tag{8}$$

We defer a proof to App. A and illustrate this behaviour in Fig. 5 for the box centre distribution $\bar{x} \sim \mathcal{N}(\mu = -1.0, \sigma = 0.5)$. There, we clearly observe that the actual super-linear growth (purple) outpaces a linear approximation (orange). While even the qualitative behaviour depends on the exact centre distribution and the input box size $\delta_x$, we can solve special cases analytically. For example, a piecewise uniform centre distribution yields quadratic growth on its support (see App. A).

Multiplying all layer-wise growth rates, we obtain the overall growth rate $\kappa = \prod_{i=1}^{L} \kappa^i$, which is exponential in network depth and super-linear in input radius. When not specifically training with the BOX relaxation, we empirically observe that the large growth factors of linear layers dominate the shrinking effect of the ReLU layers, leading to a quick exponential growth in network depth. Further, for both SABR and IBP trained networks, the super-linear growth in input radius empirically manifests as exponential behaviour (see Figs. 8 and 9). Using SABR, we thus expect the regularization induced by the robustness term to decrease super-linearly, and empirically even exponentially, with subselection ratio $\lambda$, explaining the significantly higher accuracies compared to IBP.

## 5    EVALUATION

In this section, we first compare SABR to existing certified training methods before investigating its behavior in an ablation study.

**Experimental Setup**    We implement SABR in PyTorch (Paszke et al., 2019)[1] and use MN-BAB (Ferrari et al., 2022) for certification. We conduct experiments on MNIST (LeCun et al., 2010), CIFAR-10 (Krizhevsky et al., 2009), and TINYIMAGENET (Le & Yang, 2015) for the challenging $\ell_\infty$ perturbations, using the same 7-layer convolutional architecture CNN7 as prior work (Shi et al., 2021) unless indicated otherwise (see App. C for more details). We choose similar training hyperparameters as prior work (Shi et al., 2021) and provide more detailed information in App. C.

### 5.1    MAIN RESULTS

We compare SABR to state-of-the-art certified training methods in Table 1 and Fig. 6, reporting the best results achieved with a given method on *any* architecture.

In Fig. 6, we show certified over standard accuracy (upper right-hand corner is best) and observe that SABR (◆) dominates all other methods, achieving both the highest certified and standard accuracy across all settings. As existing methods typically perform well either at large *or* small perturbation radii (see Table 1 and Fig. 6), we believe the high performance of SABR *across perturbation radii* to be particularly promising.

Methods striving to balance accuracy and regularization by bridging the gap between provable and adversarial training (✖, ●)(Balunovic & Vechev, 2020; Palma et al., 2022) perform only slightly worse than SABR at small perturbation radii (CIFAR-10 $\epsilon = 2/255$), but much worse at large radii, e.g., attaining only $27.5\%$ (✖) and $27.9\%$ (●) certifiable accuracy for CIFAR-10 $\epsilon = 8/255$ compared to $35.1\%$ (◆).

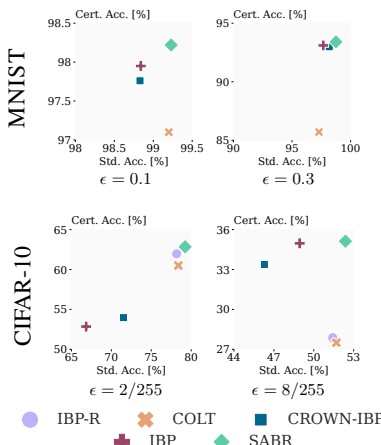

Figure 6: Certified over standard accuracy for different certified training methods. The upper right-hand corner is best.

Similarly, methods focusing purely on certified accuracy by directly optimizing over-approximations of the worst-case loss (✚, ■) (Gowal et al., 2018b; Zhang et al., 2020) tend to perform well at large perturbation radii (MNIST $\epsilon = 0.3$ and CIFAR-10 $\epsilon = 8/255$), but poorly at small perturbation radii, e.g. on CIFAR-10 at $\epsilon = 2/255$, SABR improves natural accuracy to $79.2\%$ (◆) up from $66.8\%$ (✚) and $71.5\%$ (■) and even more significantly certified accuracy to $62.8\%$ (◆) up from $52.9\%$ (✚) and $54.0\%$ (■). On the particularly challenging TINYIMAGENET, SABR again dominates all existing certified training methods, improving certified and standard accuracy by almost $3\%$.

To summarize, SABR improves strictly on all existing certified training methods across all commonly used benchmarks with relative improvements exceeding $25\%$ in some cases.

---

[1]Code released at `https://github.com/eth-sri/sabr`

Table 1: Comparison of the standard (Acc.) and certified (Cert. Acc.) accuracy for different certified training methods on the full MNIST, CIFAR-10, and TINYIMAGENET test sets. We use MN-BAB (Ferrari et al., 2022) for certification and report other results from the relevant literature.

| Dataset | $\epsilon_\infty$ | Training Method | Source | Acc. [%] | Cert. Acc. [%] |
|---|---|---|---|---|---|
| MNIST | 0.1 | COLT | Balunovic & Vechev (2020) | 99.2 | 97.1 |
| | | CROWN-IBP | Zhang et al. (2020) | 98.83 | 97.76 |
| | | IBP | Shi et al. (2021) | 98.84 | 97.95 |
| | | SABR | this work | **99.23** | **98.22** |
| | 0.3 | COLT | Balunovic & Vechev (2020) | 97.3 | 85.7 |
| | | CROWN-IBP | Zhang et al. (2020) | 98.18 | 92.98 |
| | | IBP | Shi et al. (2021) | 97.67 | 93.10 |
| | | SABR | this work | **98.75** | **93.40** |
| CIFAR-10 | 2/255 | COLT | Balunovic & Vechev (2020) | 78.4 | 60.5 |
| | | CROWN-IBP | Zhang et al. (2020) | 71.52 | 53.97 |
| | | IBP | Shi et al. (2021) | 66.84 | 52.85 |
| | | IBP-R | Palma et al. (2022) | 78.19 | 61.97 |
| | | SABR | this work | **79.24** | **62.84** |
| | 8/255 | COLT | Balunovic & Vechev (2020) | 51.7 | 27.5 |
| | | CROWN-IBP | Xu et al. (2020) | 46.29 | 33.38 |
| | | IBP | Shi et al. (2021) | 48.94 | 34.97 |
| | | IBP-R | Palma et al. (2022) | 51.43 | 27.87 |
| | | SABR | this work | **52.38** | **35.13** |
| TINYIMAGENET | 1/255 | CROWN-IBP | Shi et al. (2021) | 25.62 | 17.93 |
| | | IBP | Shi et al. (2021) | 25.92 | 17.87 |
| | | SABR | this work | **28.85** | **20.46** |

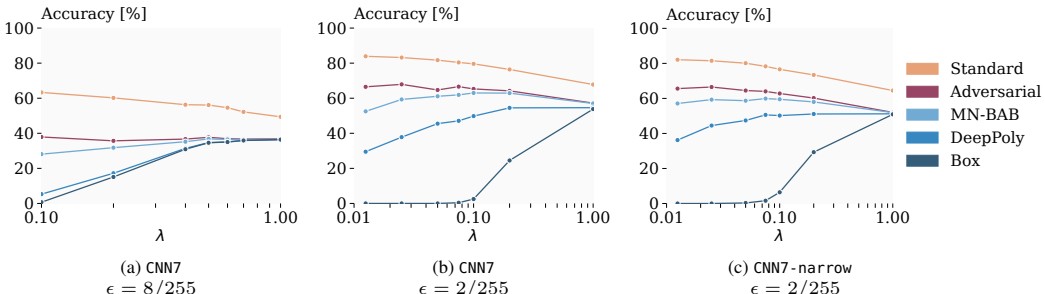

Figure 7: Standard, adversarial and certified accuracy depending on the certification method (BOX, DEEPPOLY, and MN-BAB) for the first 1000 test set samples of CIFAR-10.

In contrast to certified training methods, Zhang et al. (2022b) propose SORTNET, a generalization of recent architectures (Zhang et al., 2021; 2022a; Anil et al., 2019) with inherent $\ell_\infty$-robustness properties. While SORTNET performs well at very high perturbation magnitudes ($\epsilon = 8/255$ for CIFAR-10), it is dominated by SABR in all other settings. Further, robustness can only be obtained against one perturbation type at a time.

Table 2: Comparison of natural (Nat.) and certified (Cert.) accuracy [%] to SORTNET (Zhang et al., 2022b).

| Dataset | $\epsilon$ | SORTNET | | SABR (**ours**) | |
|---|---|---|---|---|---|
| | | Nat. | Cert. | Nat. | Cert. |
| MNIST | 0.1 | 99.01 | 98.14 | **99.23** | **98.22** |
| | 0.3 | 98.46 | 93.40 | **98.75** | **93.40** |
| CIFAR-10 | 2/255 | 67.72 | 56.94 | **79.24** | **62.84** |
| | 8/255 | **54.84** | **40.39** | 52.38 | 35.13 |
| TINYIMAGENET | 1/255 | 25.69 | 18.18 | **28.85** | **20.46** |

## 5.2 ABLATION STUDIES

**Certification Method and Propagation Region Size** To analyze the interaction between the precision of the certification method and the size of the propagation region, we train a range of models with subselection ratios $\lambda$ varying from $0.0125$ to $1.0$ and analyze them with verification methods of increasing precision (BOX, DEEPPOLY, MN-BAB). Further, we compute adversarial accuracies using a 50-step PGD attack (Madry et al., 2018) with 5 random restarts and the targeted logit margin loss (Carlini & Wagner, 2017). We illustrate results in Fig. 7 and observe that standard and

adversarial accuracies increase with decreasing $\lambda$, as regularization decreases. For $\lambda = 1$, i.e., IBP training, we observe little difference between the verification methods. However, as we decrease $\lambda$, the BOX verified accuracy decreases quickly, despite BOX relaxations being used during training. In contrast, using the most precise method, MN-BAB, we initially observe increasing certified accuracies, as the reduced regularization yields more accurate networks, before the level of regularization becomes insufficient for certification. While DEEPPOLY loses precision less quickly than BOX, it can not benefit from more accurate networks. This indicates that the increased accuracy, enabled by the reduced regularization, may rely on complex neuron interactions, only captured by MN-BAB. These trends hold across perturbation magnitudes (Figs. 7a and 7b) and become even more pronounced for narrower networks (Fig. 7c), which are more easily over-regularized.

This qualitatively different behavior depending on the precision of the certification method highlights the importance of recent advances in neural network verification for certified training. Even more importantly, these results clearly show that provably robust networks do not necessarily require the level of regularization introduced by IBP training.

**Loss Analysis** In Fig. 8, we compare the robust loss of a SABR and an IBP trained network across different propagation region sizes (all centred around the original sample) depending on the bound propagation method used. We first observe that, when propagating the full input region ($\lambda = 1$), the SABR trained network yields a much higher robust loss than the IBP trained one. However, when comparing the respective training subselection ratios, $\lambda = 0.05$ for SABR and $\lambda = 1.0$ for IBP, SABR yields significantly smaller training losses. Even more importantly, the difference between robust and standard loss is significantly lower, which, recalling §4, directly corresponds to a reduced regularization for robustness and allows the SABR trained network to reach a much lower standard loss. Finally, we observe the losses to clearly grow super-linearly with increasing propagation

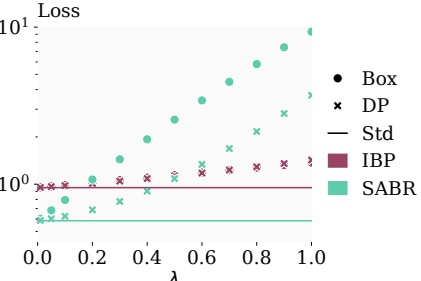

Figure 8: Standard (Std.) and robust cross-entropy loss, computed with BOX (Box) and DEEPPOLY (DP) for an IBP and SABR trained network over evaluation subselection ratios $\lambda$.

region sizes (note the logarithmic scaling of the y-axis) when using the BOX relaxation, agreeing well with our theoretical results in §4. While the more precise DEEPPOLY (DP) bounds yield significantly reduced robust losses for the SABR trained network, the IBP trained network does not benefit at all, again highlighting its over-regularization. See App. C for extended results.

**Gradient Alignment** To analyze whether SABR training is actually more aligned with standard accuracy and empirical robustness, as indicated by our theory in §4, we conduct the following experiment for CIFAR-10 and $\epsilon = 2/255$: We train one network using SABR with $\lambda = 0.05$ and one with IBP, corresponding to $\lambda = 1.0$. For both, we now compute the gradients $\nabla_\theta$ of their respective robust training losses $\mathcal{L}_{\text{rob}}$ and the cross-entropy loss $\mathcal{L}_{\text{CE}}$ applied to unperturbed (Std.) and adversarial (Adv.) samples. We then report the mean cosine similarity between these gradients across the whole test set in Table 3. We clearly observe that the SABR loss is much better aligned with both the cross-entropy loss of unperturbed and adversarial samples, corresponding to standard accuracy and empirical robustness, respectively.

Table 3: Cosine similarity between $\nabla_\theta \mathcal{L}_{\text{rob}}$ for IBP and SABR and $\nabla_\theta \mathcal{L}_{\text{CE}}$ for adversarial (Adv.) and unperturbed (Std.) examples.

| Loss | IBP | SABR |
|------|-----|------|
| Std. | 0.5586 | **0.8071** |
| Adv. | 0.8047 | **0.9062** |

**ReLU Activation States** The portion of ReLU activations which are (stably) active, inactive, or unstable has been identified as an important characteristic of certifiably trained networks (Shi et al., 2021). We evaluate these metrics for IBP, SABR, and adversarially (PGD) trained networks on CIFAR-10 at $\epsilon = 2/255$, using the BOX relaxation to compute intermediate bounds, and report the average over all layers and test set samples in Table 4. We observe that, when evaluated on concrete

Table 4: Average percentage of active, inactive, and unstable ReLUs for concrete points and boxes depending on training method.

| Method | Point | | Whole Region | | |
| | Act | Inact | Unst | Act | Inact |
|--------|-----|-------|------|-----|-------|
| IBP | 26.2 | 73.8 | 1.18 | 25.6 | 73.2 |
| SABR | 35.9 | 64.1 | 3.67 | 34.3 | 62.0 |
| PGD | 36.5 | 63.5 | 65.5 | 15.2 | 19.3 |

points, the SABR trained network has around 37% more active ReLUs than the IBP trained one and almost as many as the PGD trained one, indicating a significantly smaller level of regularization. While the SABR trained network has around 3-times as many unstable ReLUs as the IBP trained network, when evaluated on the whole input region, it has 20-times fewer than the PGD trained one, highlighting the improved certifiability.

## 6 RELATED WORK

**Verification Methods** Deterministic verification methods analyse a given network by using abstract interpretation (Gehr et al., 2018; Singh et al., 2018; 2019a), or translating the verification into an optimization problem which they then solve using linear programming (LP) (Palma et al., 2021; Müller et al., 2022; Wang et al., 2021; Zhang et al., 2022c), mixed integer linear programming (MILP) (Tjeng et al., 2019; Singh et al., 2019b), or semidefinite programming (SDP) (Raghunathan et al., 2018; Dathathri et al., 2020). However, as neural network verification is generally NP-complete (Katz et al., 2017), many of these methods trade precision for scalability, yielding so-called *incomplete* certification methods, which might fail to prove robustness even when it holds. In this work, we analyze our SABR trained networks with deterministic methods.

**Certified Training** DIFFAI (Mirman et al., 2018) and IBP (Gowal et al., 2018b) minimize a sound over-approximation of the worst-case loss computed using the BOX relaxation. Wong et al. (2018) instead use the DEEPZ relaxation (Singh et al., 2018), approximated using Cauchy random matrices. Wong & Kolter (2018) compute worst-case losses by back-substituting linear bounds using fixed relaxations. CROWN-IBP (Zhang et al., 2020) uses a similar back-substitution approach but leverages minimal area relaxations introduced by Zhang et al. (2018) and Singh et al. (2019a) to bound the worst-case loss while computing intermediate bounds using the less precise but much faster BOX relaxation. Shi et al. (2021) show that they can obtain the same accuracies with much shorter training schedules by combining IBP training with a special initialization. COLT (Balunovic & Vechev, 2020) combines propagation using the DEEPZ relaxation with adversarial search. IBP-R (Palma et al., 2022) combines adversarial training with much larger perturbation radii and a ReLU-stability regularization based on the BOX relaxation. We compare favorably to all (recent) methods above in our experimental evaluation (see §5). Müller et al. (2021) combine certifiable and accurate networks to allow for more efficient trade-offs between robustness and accuracy.

The idea of propagating subsets of the adversarial input region has been explored in the settings of adversarial patches (Chiang et al., 2020) and geometric perturbations (Balunovic et al., 2019), where the number of subsets required to cover the whole region is linear or constant in the input dimensionality. However, these methods are not applicable to the $\ell_p$-perturbation setting, we consider, where this scaling is exponential.

**Robustness by Construction** Li et al. (2019), Lécuyer et al. (2019), and Cohen et al. (2019) construct locally Lipschitz classifiers by introducing randomness into the inference process, allowing them to derive probabilistic robustness guarantees. Extended in a variety of ways (Salman et al., 2019; Yang et al., 2020), these methods can obtain strong robustness guarantees with high probability (Salman et al., 2019) at the cost of significantly (100x) increased runtime during inference. We focus our comparison on deterministic methods. Zhang et al. (2021) propose a novel architecture, which inherently exhibits $\ell_\infty$-Lipschitzness properties, allowing them to efficiently derive corresponding robustness guarantees. Zhang et al. (2022a) build on this work by improving the challenging training process. Finally, Zhang et al. (2022b) generalize this concept in SORTNET.

## 7 CONCLUSION

We introduced a novel certified training method called SABR (**S**mall **A**dversarial **B**ounding **R**egions) based on the key insight, that propagating small but carefully selected subsets of the input region combines small approximation errors and thus regularization with well-behaved optimization problems. This allows SABR trained networks to outperform *all* existing certified training methods on *all* commonly used benchmarks in terms of *both* standard and certified accuracy. Even more importantly, SABR lays the foundation for a new class of certified training methods promising to alleviate the robustness-accuracy trade-off and enable the training of networks that are both accurate and certifiably robust.

## 8 ETHICS STATEMENT

As SABR improves both certified and standard accuracy compared to existing approaches, it could help make real-world AI systems more robust to both malicious and random interference. Thus any positive and negative societal effects these systems have could be amplified. Further, while we achieve state-of-the-art results on all considered benchmark problems, this does not (necessarily) indicate sufficient robustness for safety-critical real-world applications, but could give practitioners a false sense of security when using SABR trained models.

## 9 REPRODUCIBILITY STATEMENT

We publish our code, all trained models, and detailed instructions on how to reproduce our results at `https://github.com/eth-sri/sabr`, providing an anonymized version to the reviewers. Further, we provide proofs for our theoretical contributions in App. A and a detailed description of all hyperparameter choices as well as a discussion of the used data sets including all preprocessing steps in App. C.

### ACKNOWLEDGEMENTS

We would like to thank our anonymous reviewers for their constructive comments and insightful questions.

This work has been done as part of the EU grant ELSA (European Lighthouse on Secure and Safe AI, grant agreement no. 101070617) and the SERI grant SAFEAI (Certified Safe, Fair and Robust Artificial Intelligence, contract no. MB22.00088). Views and opinions expressed are however those of the authors only and do not necessarily reflect those of the European Union or European Commission. Neither the European Union nor the European Commission can be held responsible for them.

The work has received funding from the Swiss State Secretariat for Education, Research and Innovation (SERI).

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
