# OpenReview forum: "Certified Training: Small Boxes are All You Need"
_ICLR.cc/2023/Conference — ICLR 2023 notable top 25%_

### Official Review · Reviewer_1PLc · 2022-10-15

**Confidence:** 3
**Correctness:** 3
**Technical Novelty And Significance:** 3
**Empirical Novelty And Significance:** 3
**Recommendation:** 8

**Clarity, Quality, Novelty And Reproducibility:**

The paper is mostly written clearly and is easy to follow, despite some imprecise or misleading language in certain places (see above).

As far as I know, the method is novel.

The authors claim in the paper that they would provide anonymized code to reviewers, but as far as I can tell, they did not (in appendix nor via link). Therefore, it is hard to fully judge the paper's reproducibility.

**Strength And Weaknesses:**

The paper has a clear motivation and is mostly well-written. The empirical results are good and the authors provide useful insights into why their method works well. The illustrations of IBP and SABR are very intuitive and helpful.

However, their final claim is much too strong ("Even more importantly, SABR lays the foundation for a new class of certified training methods promising to overcome the robustness-accuracy trade-off and enabling the training of networks that are both accurate and certifiably robust."). While it is true that their paper eases this tension, there is no reason to believe that this will lead to the elimination of the robustness-accuracy trade-off. Maybe it could be argued it would reduce the gap between certified-robustness and robustness.

The last sentence in the 2nd to last paragraph on page 3 ("Propagating a small region centred around the thus obtained adversarial example will then also capture the point inducing the actual worst-case loss.") is stated without justification. I am not aware of any work showing that this is true. If it exists, it should be cited. If it doesn't, then such a claim would require significant evidence.

I think the authors should be slightly more clear with the fact that the SABR loss does not provide an actual upper bound on the loss, i.e. it can only be used at train time. It would be quite helpful if they provided algorithm boxes for both train and test time in the appendix.

Minor points:
- I believe the $\epsilon$ in Figure 3 is drawn incorrectly - it should be the box's radius and not its diameter.
- In Figure 3, there is an $x_0^*$ that is never mentioned in the text.
- On page 5, the authors mention the typical magnitudes for growth rates in linear layers. Surely, these depend on how the model is trained? They should clarify if this holds for all training methods or only for theirs or what.
- Figure 6 is not great. Why are there no ticks on the x and y axes?

**Summary Of The Paper:**

The authors propose a way to simultaneously increase both the certified and clean accuracy of certifiable networks by reducing the regularization that IBP training imposes at train time. They provide theoretical and empirical insights into why their method leads to good performance.

**Summary Of The Review:**

I think the paper presents a well-motivated method with strong results. If the authors slightly improve their presentation, I believe this paper will be of interest to the ICLR community.

---

> ### Author Response · Authors · 2022-11-13
> **Response to Reviewer 1PLc**
>
> $\newcommand{Rf}{\textcolor{orange}{1PLc}}$
> We thank the reviewer $\Rf$ for their helpful comments and insightful suggestions. We are particularly glad that they appreciate both our empirical and theoretical results as well as their presentation. Below we address their remaining concerns.
>
> **Q: The claim of laying the foundations for a new class of methods promising to overcome the robustness-accuracy trade-off seems rather strong.**
> We agree with the reviewer that this statement might be over-optimistic with respect to future work and have toned it down (see also the updated abstract in the PDF).
>
> **Q: Can you discuss the claim that propagating a small region around the PGD point will capture the worst-case loss?**
> This statement was intended to be part of the intuition given in the preceding lines, but we agree that it can be misunderstood and have updated the section to remove this claim.
> We note that this intuition generally holds for convex optimization problems with suitably chosen step-sizes, we agree that this is not the case for the highly non-convex neural networks we consider.
>
> **Q: Can you point out more clearly that the SABR loss does not provide an actual upper bound on the loss and provide algorithms for train and test time?**
> We want to point out that we stated this explicitly in the first paragraph of page four, but have updated Section 3 to highlight this point more prominently.
>
> **Minor points:**
> * Thanks for the pointer regarding $\epsilon$ (and $\tau$) in Figure 3, we have corrected it.
> $x_0^*$ in Figure 3 is the random initialization of the PGD attack. We have removed the labeling to not overload the figure.
> * Growth rates do indeed depend on the training method. The quoted range of 10-100 applies to adversarially trained networks. We have added Table 10 to Appendix D.4 showing the per-layer growth rates depending on the training method..
> * We have updated Figure 6 to include labeled axes.
> * The code was provided via a PC comment which unfortunately was only made after the review phase began. We invite the reviewer to have a look at the provided code.
> * We added two algorithm boxes for training to Appendix D.5. As MN-BaB is fairly complex and used as a black box in this work, we refer to Ferrari et al. [1] for more details and suggest their blog post (https://www.sri.inf.ethz.ch/blog/mnbab) for an intuition.
> We hope to have been able to address the reviewer's concerns, are happy to answer any follow-up questions, and look forward to their reply.
>
> **References**
> [1] Ferrari, Claudio, et al. "Complete verification via multi-neuron relaxation guided branch-and-bound." ICLR 2022

---

> > ### Comment · Reviewer_1PLc · 2022-11-15
> > **Response to rebuttal**
> >
> > I acknowledge the authors' rebuttal. They have adequately addressed all my concerns and I have updated my score accordingly.
> >
> > I continue to support this paper's acceptance.

---

### Official Review · Reviewer_Bdkn · 2022-10-17

**Confidence:** 4
**Correctness:** 3
**Technical Novelty And Significance:** 3
**Empirical Novelty And Significance:** 2
**Recommendation:** 6

**Clarity, Quality, Novelty And Reproducibility:**

- Clarity and quality: The paper is mostly clear and well written.
- Novelty: Using small boxes is novel
- Reproducibility: Code is provided.


**Strength And Weaknesses:**

Strength:
- This work proposes to combine adversarial attacks and box propagation / interval bound propagation such that the boxes used in box or interval bound propagation can be small.
- There is a theoretical analysis on the growth of boxes during interval bound propagation. The derivation has considered the imbalance between active and inactive ReLU neurons, compared to a previous work assuming symmetric input to ReLU activations at initialization only.
- Compared to existing IBP-based works, this work has consistent improvement across different eps settings.

Weaknesses:

- Although neither of IBP-R and IBP (Shi et al.’s version) achieve comparable performance on 2/255 and 8/255 at the same time compared to this work, this work relies on a careful choice of $\ell_1$ and $\lambda$ hyperparameters for each setting respectively (with values like 0.4, 0.6, 0.1, 0.7 in Table 5), while I don’t see a principled mechanism for setting these hyperparameters. It is unclear how sensitive the performance is w.r.t. different choice of hyperparameters.
- MN-BaB (a much stronger verifier compared to IBP) is used for evaluating the models trained by this work, while previous works such as CROWN-IBP and IBP simply use IBP. Different methods in Table 1 were evaluated using different verifiers, and thus the comparison is not fair. This work spent a long time on verification (28h on CIFAR-10), while previous works using IBP at test time should be able to finish within a minute. It is unclear what the performance of previous works would be, if MN-BaB were used for those previous works (especially for CIFAR eps=2/255).
- Compared to both IBP and IBP-R together, the improvement of the proposed work is marginal on robustness (on CIFAR 2/255, IBP-R has 78.19/61.97 clean/robust accuracy v.s. 79.52/62.57 by this work; on CIFAR 8/255, IBP (Shi et al.’s version) has 48.94/34.97 v.s. 52.00/35.25 by this work).



**Summary Of The Paper:**

This work tries to improve certified training based on box propagation (DiffAI) / interval bound propagation (IBP). The proposed method tries to find a point with a high loss within the perturbation region by adversarial attack. Then box propagation is performed around the high-loss input point with a small box only, in contrast to the whole perturbation region used in previous IBP works. The goal of the modification is to improve both clean accuracy and certified robust accuracy.

**Summary Of The Review:**

The method is sound and promising, and this work has demonstrated some improvement compared to previous works:
- Better clean accuracy
- More consistent performance across different perturbations.

However, experiments are not convincing enough:
- Improvement on robustness is not quite significant
- Different methods are evaluated using verifiers with significantly different strengths
- Slow verification at test time
- A lack of discussion on the choice of hyperparameters.

---

> ### Author Response · Authors · 2022-11-13
> **Response to Reviewer Bdkn**
>
> $\newcommand{Rtr}{\textcolor{blue}{Bdkn}}$
> We thank reviewer $\Rtr$ for their insightful comments and are encouraged that they consider our method novel and promising. Below we address their remaining concerns:
>
> **Q: MN-BaB is a much stronger verifier than IBP. Can you analyze previous works with MN-BaB, especially for CIFAR10 at eps=2/255?**
> We first want to point out that both COLT and IBP-R use similarly powerful verifiers with even longer runtimes (MILP and beta-CROWN, respectively). We have run MN-BaB using the same settings as for our networks on all CIFAR10 settings where networks were published (2/255 and 8/255 for CROWN-IBP (2 different checkpoints each from Zhang et al.),  2/255 and 8/255 for COLT, and 8/255 only for IBP from Shi et al.). At 8/255, the IBP trained network reaches $35.3$\% certified accuracy (almost identical to ours), while still only achieving $48.6$\% compared to our $52.0$\% natural accuracy. While improvements at 2/255 are more significant for the CROWN-IBP trained network, it only reaches $58.2$\% certified accuracy compared to our $62.6$\% which also comes with a much higher ($80.3$\% vs $71.5$\%) natural accuracy. See below for the full results, which we added to Table 7 in the new Appendix D.1.
>
> | $\epsilon_\infty$ | Training Method | Acc. [\%]      | Cert. Acc. [\%] |
> |-------------------|-----------------|-------------------|-----------------|
> | 2/255             |  COLT           | 78.4         | 61.0       |
> | 2/255             | CROWN-IBP   | 71.3        | 58.2        |
> | 2/255             | SABR           | **79.5**   | **62.6**  |
> ||||
> | 8/255             | COLT          | 51.7          | 27.6       |
> | 8/255             | CROWN-IBP  | 45.4          | 33.2       |
> | 8/255             | IBP       | 48.9         | **35.3**  |
> | 8/255             | SABR           | **52.0**   | **35.3**   |
>
> **Q: Can you discuss to what extent SABR's better performance compared to IBP and IBP-R is due to hyper-parameter choices?**
> First, we want to highlight that we show the sensitivity to the most important hyperparameter, the subselection ratios $\lambda$, in Figure 7. There, we observe flat maxima, indicating low sensitivity over wide parameter ranges. We note that we selected hyperparameters via coarse manual search, generally selecting larger $\lambda$ for larger $\epsilon$ as stronger regularization.
> We point out that other state-of-the-art certified training methods often tune multiple hyper-parameters (e.g. $\alpha$, $\kappa$, and $\rho$ for IBP-R), sometimes even using commercial tools (e.g. SigOpt for COLT), or select the checkpoint with the best test-set performance (Crown-IBP). Therefore, we believe that manually selecting one major hyperparameter ($\lambda$) and the $\ell_1$ regularization constitutes a fair comparison. We actually believe that having so few hyperparameters can be seen as an advantage of our method.
>
> **Q: Can you discuss SABR’s performance compared to the best of IBP and IBP-R (especially for CIFAR10)?**
> When combining the better certified and better natural accuracy from IBP and IBP-R at any setpoint, they indeed reach similar performance to SABR. However, as certified training is dominated by a robustness/accuracy trade-off, we believe that it is quite remarkable for any method/network to beat the best combination of two quite different approaches. In fact, we believe this significant improvement of the robustness/accuracy Pareto frontier to be one of SABR's strengths. Further, in some settings (e.g. TinyImageNet), SABR significantly outperforms all current state-of-the-art certified defenses in terms of both standard and certified accuracy.
>
> **Q: Is verification with MN-BaB unreasonably slow?**
> Usually, models are certified offline, making an increased verification time a one-time cost and thus generally less problematic than inference time penalties. Further, both IBP-R and COLT use even more expensive verification approaches, taking >27 days (273s per property which did not time out times at least one property per sample) and 2 days, respectively.
> As verification times of multiple minutes per sample are common in the neural network verification community [1], we believe the settings we chose for MN-BaB and the resulting verification times to be quite reasonable. Finally, we note that most samples can actually be verified quickly (for Cifar10 8/255 the median time for successful verification is e.g. 0.12s with a mean of 6.7s) however for samples that can be neither verified nor falsified, MN-BaB attempts verification until a timeout of several hundred seconds is reached, leading to a significantly increased total runtime for verification.
>
> We hope to have been able to address the reviewer’s concerns, are happy to answer any follow-up questions, and are looking forward to their response.
>
> **Resources:**
> [1] Bak et al. "The second international verification of neural networks competition (vnn-comp 2021): Summary and results." arXiv 2021

---

> > ### Comment · Reviewer_Bdkn · 2022-11-15
> > **Post-rebuttal update**
> >
> > Thanks to the authors for the response. The experiments have become more complete given the new evaluation using MN-BaB for all the methods. I think there remain some weaknesses on the significance of improvement, as the proposed method does not consistently outperform existing works on both natural accuracy and verified accuracy across various perturbation radii, and both natural accuracy and verified accuracy are below the results by Linf-net on CIFAR 8/255. Besides, the proposed method comes at the cost of slow verification compared to IBP and Linf net. While the authors say “Usually, models are certified offline…”, I do not fully agree with this argument, since we can only obtain certification on the visited instances, while we may also want to certify new instances online. However, the method of using small boxes sounds interesting and may inspire future works, and there has been satisfactory improvement on several settings. Therefore, I am raising my recommendation to 6.

---

### Official Review · Reviewer_6kss · 2022-10-23

**Confidence:** 4
**Correctness:** 4
**Technical Novelty And Significance:** 4
**Empirical Novelty And Significance:** 4
**Recommendation:** 8

**Clarity, Quality, Novelty And Reproducibility:**

### Quality
The idea behind SABR is straightforward, giving me the confidence that everything described is actually required to make things work. In addition, the SABR can be used with any combination of heuristic attack and propagation method; this enables future advances in the field (e.g. better heuristic attacks; more efficient propagation methods) to also improve the results from SABR.

Some suggestions and questions:

- Suggestion: I'd love to see an ablation study for the methodology described in Section 3. How would SABR perform if we ...
  - Used a weaker heuristic attack (e.g. FGSM)
  - Used a ball of size $\lambda \epsilon$ around the original input.
  - Used the ball around the preliminary centre x*
  - Used a different propagation method (does the 'well-behaved optimization' intuition from Jovanovic et al. still hold up here?)
  - Having said that, I understand that this all takes time, and I think this is already an excellent paper without that.


### Clarity
The attention to detail paid to the exposition of the paper (textual flow, choice of variables in equations, thoughtful use of color and symbols across text and figures) is remarkable, and makes understanding the contributions of the paper easy (which is how it should be)! Thanks for putting in the effort.

Some suggestions and questions:

- Suggestion: On Page 1, the paper states that “SABR, thus, achieves state-of-the-art standard and certified accuracies ... across all commonly used settings”. A casual reader might think this means that SABR achieves state-of-the-art standard accuracy as compared to neural networks _in general_ (or at least state-of-the-art standard accuracy as compared to neural networks with robustness claims that are not certified, like adversarially trained networks). More precise language would be helpful here. Some possibilities depending on what is true:
  - "SABR achieves state-of-the-art standard accuracy among all networks with non-trivial certifiable accuracy."
  - "SABR is on the Pareto frontier of standard and certified accuracy."
- Q: I'm unfamiliar with the formula for cross-entropy loss shown in Eq. 2 and Eq. 5. Would you point me to a reference? (Ok to not spend your rebuttal space explaining this to me ...)
- Minor Q: On Page 8, the paper states that "When propagating the full input region, the SABR trained network yields a much higher robust loss". Wouldn't the values be equal at $\lambda = 1$
- Minor Q: The paper is not clear on what happens if an adversarial example (i.e. one that is misclassified) is not found via PGD. Do we select an arbitrary propagation region, or do we simply select the x' that is closest to being misclassified (i.e. that with the smallest difference in logits between the correct label and the next label)
- Minor suggestion: Remembering which shade of blue corresponded to which accuracy was challenging. Maybe consider having standard / adversarial / certified be in different hues, with the three certified accuracies having the same hue and different luminances? (This is obviously up to you.)

### Novelty
The results presented represent the state-of-the-art in comparison to all existing non-probabilistic techniques. However, Salman et al. [1] seem to claim a certifiable and standard accuracy on $l_\infty$ CIFAR-10 networks of 68.2% and 86.2% respectively [2]. This is significantly better than the results reported in Table 1 (62.57% / 79.52%). Does this work explicitly exclude comparisons against randomized smoothing techniques (because of the probabilistic nature of the claim?) If so, that should be made clear in the abstract & introduction.

[1]: Salman, Hadi, et al. "Provably robust deep learning via adversarially trained smoothed classifiers." Advances in Neural Information Processing Systems 32 (2019). https://arxiv.org/pdf/1906.04584.pdf
[2]: See bottom of Page 8 for how they derive this from $l_2$ robustness.

### Reproducibility
The paper clearly describes details required to reproduce their results. (For example, the PGD attack used in their ablation study is clearly specified; no wondering on my part "how strong of a PGD attack did they use??)

**Strength And Weaknesses:**

See the "Clarity, Quality, Novelty And Reproducibility" section below for strengths and weaknesses split by category.

**Summary Of The Paper:**

The paper proposes a novel training method (SABR) that creates models balancing standard and certifiable accuracy. Conceptually, SABR combines the best of adversarial training (using strong heuristic attacks as a lower bound on the worst-case loss) and certified training (using overapproximations of the adversarial output region as an upper bound on the worst-case loss). In practice, SABR enables us to retain the high accuracy provided by adversarial training while producing networks that have a high certifiable accuracy using existing verification methods.

SABR uses a fast heuristic method (e.g. PGD) to find an adversarial example $x'$ within $\epsilon$ of the input $x$. Rather than propagating the entire input ball of radius $\epsilon$ to the output, SABR selects a ball of radius $\lambda \epsilon$ that 1) is contained within the original ball and 2) contains $x'$. This approach guarantees that the overapproximation from SABR 1) is contained in the overapproximation of the full adversarial input region while 2) likely contains regions of the output that are highly adversarial. While there are no theoretical guarantees that such an approach should provide any certifiable accuracy, the work demonstrates that SABR is on the Pareto frontier for standard and certifiable accuracy on $l_\infty$ networks.

**Summary Of The Review:**

The paper presents a simple idea with a sound basis and strong empirical results which the field will be able to build on. The clarity and reproducibility of the paper is outstanding and represents a clear acceptance ( the failure to compare against results from randomized smoothing notwithstanding).

---

> ### Author Response · Authors · 2022-11-13
> **Response to Reviewer 6kss**
>
> $\newcommand{Rt}{\textcolor{green}{6kss}}$
> We thank reviewer $\Rt$ for the detailed review, helpful suggestions, and interesting questions. We are encouraged to hear that they appreciate the presentation, clarity, novelty, and importance of our work. Below we address their comments and suggestions:
>
> **Q: Can you add an additional ablation study analyzing the different components of SABR?**
> Great suggestion! We have added an ablation study on CIFAR10 at 2/255 including the settings suggested by the reviewer as well as choosing center locations $x’$ randomly. We report results in Table 8 in Appendix D.2 (and below). Results are generally as expected: Reducing regularization by choosing propagation region centers as the original input or at random increases standard accuracy slightly at the cost of significantly reduced certified accuracy. Using a weaker attack such as FGSM instead of PGD to determine the propagation region centre has a similar but less pronounced effect. Increasing regularization by using the preliminary centre $x^*$ as the center slightly increases certified accuracy at the cost of reduced standard accuracy (for larger $\epsilon$ it becomes strictly worse, however). Using the more precise CROWN-IBP instead of IBP propagation during training leads to decreased standard and certified accuracy. This agrees well with results suggesting that more precise methods induce harder optimization problems [1]. See the full results below
>
> | Training Method | Acc. [\%] | Cert. Acc. [\%] |
> |-----------------------|--------------|-----------------|
> | SABR                 | 80.4         | 61.0            |
> | + centered          | 82.5         | 27.4            |
> | + random            | 84.1         | 28.3            |
> | + FGSM              | 80.8         | 58.2            |
> | + no projection    | 80.0         | 61.9            |
> | + CROWN-IBP    | 80.3         | 56.7            |
>
> Upon publication, we will move these results forward to the main experimental results using the extra page of the camera-ready version.
>
> **Q: Can you clarify that SABR only obtains SOTA accuracies for certifiably trained networks?**
> Thanks for pointing out that our formulation can be misunderstood. We have updated it to read “SABR [...] improves on state-of-the-art certified defenses in terms of standard and certified accuracies”.
>
> **Q: Can you provide a derivation for the formulation of the cross-entropy loss shown in Eq. 2 and Eq. 5?**
> We have added a derivation of this result in Appendix B.1.
>
> **Q: Should the loss for the SABR- and IBP-trained network not be equal at $\lambda=1$?**
> In Figure 8 the SABR network is always trained with the same $\lambda_\text{train}$ and then evaluated with different propagation region sizes $\lambda_\text{eval}$. This is in contrast to Figure 7, where the x-axis is $\lambda_\text{train}$. We made this distinction clearer.
>
> **Q: What happens if we do not find an adversarial example via PGD during training?**
> We always select the sample with the highest cross-entropy loss found during PGD, irrespective of whether it leads to misclassification. We made this more clear in the paragraph “Selecting the Propagation Region Size”.
>
> **Q: Can you choose a more distinctive color scheme for Figure 7?**
> We thank the reviewer for their feedback and have updated the color scheme.
>
> **Q: Can you make the (lack of) comparison to probabilistic models clearer in abstract and intro?**
> Yes. As the reviewer suggests, the main reason for not comparing to probabilistic guarantees is their probabilistic nature. Additionally, probabilistic methods can not analyse a model as is, but construct new classifiers with significant runtime penalties at inference time (factor of n=100 or more see Figure 8 in Salman et al. [2]) compared to already much larger base models. Due to these fundamental differences, we do not compare to probabilistic methods. We have made this clearer in the related work, intro, and abstract (see the version in the updated PDF).
>
> We hope to have addressed the reviewer's comments and are happy to answer any further questions.
>
> **Resources:**
> [1]: Jovanović et al. "On the Paradox of Certified Training.", TMLR 2022 (formerly “Certified defenses: Why tighter relaxations may hurt training?”)
> [2]: Salman et al. "Provably robust deep learning via adversarially trained smoothed classifiers." Neurips 2019

---

> > ### Comment · Reviewer_6kss · 2022-11-14
> > **Thanks for the update**
> >
> > Thank you for your update! This was an excellent paper before, and I appreciate the effort the authors put in to improve it (especially within the tight constraints of the review period).

---

> > > ### Author Response · Authors · 2022-11-15
> > > **Fixed Typo and Updated PDF**
> > >
> > > We are happy to hear that we could address all the reviewers' suggestions and have fixed the typo in the updated PDF; Thank you for spotting it!
> > >
> > > We are happy to address any further follow-up questions and, in light of the very positive review, are hopeful the reviewer may strongly recommend this work for acceptance.

---

> > ### Comment · Reviewer_6kss · 2022-11-15
> > **Kudos on using colors to distinguish the reviewers**
> >
> > That made it a lot easier to track which reviewer the authors were responding to.

---

> > ### Comment · Reviewer_6kss · 2022-11-15
> > **nit: typo in cross-entropy loss derivation in app b.1?**
> >
> > The indicator function in the second line of the derivation is $1_{i=y}$; should it be $1_{i=t}$? The derivation looks correct to me otherwise.

---

### Official Review · Reviewer_mQ2H · 2022-10-25

**Confidence:** 4
**Correctness:** 4
**Technical Novelty And Significance:** 4
**Empirical Novelty And Significance:** 4
**Recommendation:** 8

**Clarity, Quality, Novelty And Reproducibility:**

Clarity and quality are very good.

- Abstract
  - I strongly recommend that the authors refrain from prematurely framing their contribution as novel, as early as the fourth word in the abstract.
    - Such statements are extremely likely to end up harming the contribution. As other reviewers point out, the picture is not as clear cut with respect to randomized smoothing methods.
    - I would also recommend that the problem and method are described first, before how it compares to current SOTA.

- Section 1
  - What was meant to be conveyed through the statement "This yields networks with complex neuron interactions"? Is that about the ReLU activations? As it stands, it is rather vague and redundant.
  - Please spell out the main "commonly used settings" deferring further details to the experiments section.
- Section 3
  - First sentence: we address "this" challenge. What challenge?
  - It's not clear what the words "capture" and "actual" serve, and they appear more than once in the same paragraph.
  - I believe "often still captures the actual worst-case loss" was meant as "is a plausible approximation of the worst-case loss".
  - Expressing intuitions is okay, but there's no need to make inaccurate or unsupported claims in the process.
    - I strongly recommend that the authors rewrite the sentences following "Intuitively, often only small subsets of the input are misclassified and only a single point will realize the worst-case loss." Those statements almost surely don't hold in general, and need not be settled to communicate intuition in the first place.
  - It would help greatly to include rudimentary experiments to provide better intuitions and empirical evidence to support those motivating observations.
  - Second paragraph: we illustrate "this intuition".
  - I recommend to start a new paragraph at: "we tackle this problem", after clarifying in the first paragraph what the challenge/problem is.
  - This statement "leading to significantly reduced approximation errors and thus more precise, although not necessarily sound over-approximation of the loss" is rather too confounding.
    - Please define the propagation of the small region as a proxy or an auxiliary problem to the original problem of propagating the full region. Please include an equation similar to Eq.3 w.r.t. $B_p^{\tau_p}(x')$ or some other placeholder symbol.
    - Then, please explain why this new approximation problem admits a more precise solution, and why it can be understood as a regularization in-between the two extremes.
    - Then, please explain how it relates to the original approximation problem, clarifying what was meant by "not necessarily sound over-approximation".
  - The paragraph titled "Selecting the Propagation Region" is good enough. The leading paragraphs need not say as much, IMHO, and perhaps can be employed to better anticipate the theoretical analysis culminating in Theorem 4.1, and possibly other potential analyses going beyond BOX.

The rest of the paper looks good. Only that the repeated claim of overcoming the robustness-accuracy trade-off is too strong, and would at least requires demonstration on a range of learning problems, which the current work does not substantiate.

**Strength And Weaknesses:**

- Strengths
  - Introducing an auxiliary IBP problem admitting better approximations.
  - Theoretical analysis of hyperbox growth, with a detailed discussion of the role of ReLU activations following Shi et al. (2021).
  - Consistently improved accuracies corroborating the theory, in addition to a valuable ablation study.
- Weaknesses
  - The presentation deviates from professional technical writing in a number of critical parts. This leads to unnecessary confusion, as well as a couple inaccurate or unsubstantiated claims:
    - Improving on all SOTA methods
    - Promising to overcome the robustness-accuracy trade-off

**Summary Of The Paper:**

The authors propose a variant of interval bound propagation (IBP) for certified training using a small region around a preliminary center, e.g. from a PGD attack. Following a theoretical analysis of the employed box propagation, paying special attention to the role of ReLU activations, the authors present a set of experiments showing consistent improvements in terms of both the standard and robust accuracy, along with ablation studies concerning a number of related considerations.

**Summary Of The Review:**

~~I'm assigning a score of 8 since 7 is no longer available.~~
The revised manuscript addressed my reservations.

---

> ### Author Response · Authors · 2022-11-13
> **Response to Reviewer mQ2H**
>
> $\newcommand{Ro}{\textcolor{purple}{mQ2H}}$
> We thank reviewer $\Ro$ for their detailed review, helpful suggestions, and insightful comments. We are particularly glad to hear that they appreciate the clarity, quality, and novelty of our work. We have incorporated the reviewer’s suggestions on how to improve the presentation of our work (and especially section 3) even further (see also the updated abstract in the PDF) and answer the remaining questions below.
>
> **Q: Can you provide some simple experiments supporting the provided intuitions?**
> Great suggestion! We trained a small convolutional network on MNIST $\epsilon=0.1$ using SABR and computed the exact worst-case maximum margin loss using mixed integer linear programming (MILP) for the first 100 test set samples. We observe that the differences between these exact bounds and those computed with SABR have a smaller mean, variance, and mean absolute value compared to both PGD (using a significantly stronger attack than for SABR) and IBP bounds (see Table below). This suggests that the SABR loss (Eq. 5) is indeed a better proxy for the worst-case loss. For (significantly) more detail, please see the new Appendix D.3.
>
> |Bounding Method| $\mathbb{E}[\cdot]$ | $Var$ | $\mathbb{E}[\|\cdot\|]$ |
> |-----------------|-------------|--------|---------------|
> | SABR           | -0.106      | 0.041  | 0.161         |
> | IBP               | 1.182       | 0.497  | 1.182         |
> | PGD             | -0.244      | 0.049  | 0.244         |
>
> Upon publication, we will use the extra page to include these results in the main paper.
>
> **Q: What is meant by “networks with complex neuron interactions”?**
> We hypothesised that SABR-trained networks exhibit `more complex neuron interactions’ based on the observation that they benefit significantly more from using the MN-BaB instead of DeepPoly verifier compared to e.g. IBP trained networks. One of the key differences between the two verifiers is that, while DeepPoly computes convex relaxations for one ReLU activation at a time, MN-BaB uses multi-neuron constraints that capture interactions between groups of ReLU activations. We thus conclude that these interactions between neurons are critical for SABR-, but not (as much) for IBP-trained networks. As we do not have the space for this more detailed discussion in the introduction, we have removed the statement there.
>
> We hope to have addressed the reviewer's concerns, are happy to answer any further questions, and appreciate any further feedback (in particular on the implementation of the adaptations they suggested).

---

> > ### Comment · Reviewer_mQ2H · 2022-11-13
> > **Thanks for the follow up**
> >
> > Please consider pointing to the appendices in the main body.
> >
> > By all means, please feel free to communicate your intuitions, insights, and motivations. I only meant to point out that complex interactions can mean anything.

---

> > > ### Author Response · Authors · 2022-11-15
> > > **Reply to Reviewer mQ2H**
> > >
> > > We are happy we could address the reviewers' concerns and have added references to the appendices in our working version. We will update the PDF once all reviewers have replied to unnecessarily many revisions.
> > >
> > > Upon publication, we will use the extra space of the camera-ready version to communicate more intuitions and insights along with the new experiments supporting them.

---

### Author Response · Authors · 2022-11-13
**Main Response**

$\newcommand{Ro}{\textcolor{purple}{mQ2H}}$
$\newcommand{Rt}{\textcolor{green}{6kss}}$
$\newcommand{Rtr}{\textcolor{blue}{Bdkn}}$
$\newcommand{Rf}{\textcolor{orange}{1PLc}}$
We thank all reviewers for their interesting questions, insightful comments, and helpful suggestions. We are particularly encouraged that the reviewers appreciate the novelty ($\Ro$, $\Rt$, $\Rtr$, $\Rf$) and significance ($\Ro$,$\Rt$, $\Rf$) of our work as well as the quality of its exposition ($\Ro$, $\Rt$, $\Rf$) and believe it to be of interest to the (ICLR) community ($\Rt$, $\Rf$).

We did not identify any shared concerns or questions regarding the technical contributions of our work and answer the remaining questions in reviewer-individual responses.

We have incorporated the excellent suggestions on how to further improve the clarity of our presentation ($\Ro$, $\Rt$, $\Rf$), including a slightly less optimistic outlook ($\Ro$, $\Rf$), into the updated version of our paper (see also the updated abstract in the PDF).

Based on the suggestions of reviewers $\Ro$, $\Rt$, $\Rtr$, and $\Rf$, we have added additional experiments supporting the intuitions we give in Section 3 (Appendix D.3), ablating the different components of SABR (Appendix D.2), showing the effect of verification methods on other certified training methods (Appendix D.1), and the dependence of the abstraction size growth rate on the training method (Appendix D.4), respectively.

We hope to have been able to address all of the reviewers’ questions and concerns and are happy to answer follow-up questions.

---

### Public Comment · ~Bohang_Zhang1 · 2023-02-12
**Great work!**

Dear authors,

Thanks for the great work and congratulations! As a researcher in this field, it is particularly exciting to see that the deterministic certified accuracy under $\epsilon=2/255$ on CIFAR-10 can now reach such a high value!

I would also like to point out our recent work at NeurIPS22, which targets the deterministic $\ell_\infty$ certified robustness as well. We propose the theoretically inspired SortNet architecture that achieves high certified $\ell_\infty$ robustness on a variety of datasets ranging from CIFAR-10 to ImageNet64. It seems that your approach can achieve better result under small perturbation radius, while our works can achieve better result under larger radius. I think it may be great to make a discussion and comparison between the two works. Thank you!

[1] Bohang Zhang, Du Jiang, Di He, Liwei Wang. "Rethinking Lipschitz Neural Networks and Certified Robustness: A Boolean Function Perspective",  NeurIPS 2022.

---

> ### Author Response · Authors · 2023-02-14
> **Thank you!**
>
> Dear Bohang,
>
> We are happy to hear that you like our work!
>
> We saw your work at NeurIPS and think it is highly interesting that dedicated architectures seem to work so well at very large perturbation radii. We are happy to also include concurrent work in our discussion and will update it for the camera-ready version.
>
> The authors

---

### Decision · Program_Chairs · 2023-01-20

**Decision:**

Accept: notable-top-25%

**Justification For Why Not Higher Score:**

The paper achieves SOTA results on deterministic certified robustness with a simple technique. However, the authors do not develop any significant theoretical insight or outperform SOTA certified robustness methods that use randomized smoothing. Hence, I would not recommend this paper to be accepted as an oral.

**Justification For Why Not Lower Score:**

The paper achieves SOTA results on a hard problem (deterministically certified robustness to adversarial perturbations). This field has seen intense activity in recent years and being able to outperform the best techniques is indeed a significant achievement and deserves to be highlighted as a spotlight.

**Metareview: Summary, Strengths And Weaknesses:**

The authors develop a novel approach for training networks to be certifiably robust against norm-bounded adversarial attacks. The key insight developed by

Strengths:
1. The proposed technique is simple to implement and combines the strengths of adversarial training and certified training.
2. The paper achieves SOTA results relative to all prior deterministic certification/certified training methods.

Weakensses:
1. The paper does not outperform randomized smoothing as a certified defense. However, considering that randomized smoothing has other defects (only provides a probabilistic guarantee and requires significantly more inference time computation), the proposed techniques are certainly valuable. The initial version of the paper was not clear about this, but through the discussion phase the authors revised the paper to clarify this point.



**Note From Pc:**

if the above contains the word "oral" or "spotlight" please see: "oral" presentation means -> notable-top-5% and "spotlight" means -> notable-top-25%. As stated in our emails, we are disassociating presentation type from AC recommendations

**Summary Of Ac-Reviewer Meeting:**

No meeting